# TaylorGAN: Neighbor-Augmented Policy Update for Sample-Efficient Natural Language Generation

**Chun-Hsing Lin**    **Siang-Ruei Wu**    **Hung-Yi Lee**    **Yun-Nung Chen**
National Taiwan University, Taipei, Taiwan
{jsaon92, raywu0}@gmail.com    hungyilee@ntu.edu.tw    y.v.chen@ieee.org

## Abstract

Score function-based natural language generation (NLG) approaches such as RE-INFORCE, in general, suffer from low sample efficiency and training instability problems. This is mainly due to the non-differentiable nature of the discrete space sampling and thus these methods have to treat the discriminator as a black box and ignore the gradient information. To improve the sample efficiency and reduce the variance of REINFORCE, we propose a novel approach, TaylorGAN, which augments the gradient estimation by off-policy update and the first-order Taylor expansion. This approach enables us to train NLG models from scratch with smaller batch size — without maximum likelihood pre-training, and outperforms existing GAN-based methods on multiple metrics of quality and diversity.[1]

## 1 Introduction

Generative adversarial networks (GAN) [14] have advanced many applications such as image generation and unsupervised style transfer [19, 28]. Unsurprisingly, much effort has been devoted to adopting the GAN framework for unsupervised text generation [5, 9, 11, 17, 25, 32, 33]. However, in natural language generation (NLG), more challenges are concerned, such as passing discrete tokens through a non-differentiable operation, which prohibits backpropagating the gradient signal to the generator.

To address the issue about non-differentiablity, researchers and practitioners used score function-based gradient estimators such as REINFORCE to train GANs for NLG, where the discriminator is cast as a reward function for the generator. These methods suffer from poor sample efficiency, high variance, and credit assignment problems. We argue that it is disadvantageous to utilize the discriminator as a simple reward function when it is known that gradient-based backpropagation is more effective for optimization.

In this paper, we propose a novel unsupervised NLG technique, TaylorGAN, where the approximated reward of sequences improves the efficiency and accuracy of the estimator. Our contributions are 3-fold:

- This paper proposes a novel update formula for the generator to incorporate gradient information from the discriminator during training.
- The experiments demonstrate that TaylorGAN achieves state-of-the-art performance without maximum likelihood pre-training.
- Our model does not require additional variance reduction techniques used in other REINFORCE-based counterparts such as large batch size [9], value estimation model [17] and Monte-Carlo rollouts [32].

## 2 Background

GAN is an innovative approach of generative modeling. Instead of learning a probabilistic model via maximum likelihood estimation (MLE), GAN is a two-player minimax game, in which the generator $G_\theta$ aims at mapping random noises $z$ to realistic samples and the discriminator $D_\phi$ focuses on classifying whether a sample is from real data $p_{data}$ or from the generator.

$$\min_\theta \max_\phi \mathop{\mathbb{E}}_{x \sim p_{data}} [\log D_\phi(x)] + \mathop{\mathbb{E}}_{z \sim p_z} [\log(1 - D_\phi(G_\theta(z)))] \tag{1}$$

In the standard architecture of GAN, the generator's output $G_\theta(z)$ is directly connected as the input to the discriminator in a fully differentiable manner, which means that the gradients of the objective $\nabla_\theta \log(1 - D_\phi(G_\theta(z)))$ can be directly backpropagated to the generator's parameters $\theta$. However, in NLG, data are defined in a discrete domain, which is essentially *non-differentiable*. In order to avoid the intractability of gradients, text GANs proposed various approaches for estimating the gradients.

**Continuous Relaxation**  Continuous relaxation approaches such as Gumbel-Softmax [18] approximate discrete data in terms of continuous variables such as outputs of a softmax function. While it allows us to ignore the non-differentiable discrete sampling [21, 25], several issues may be occurred.

First of all, the discriminator only needs to spot the difference between the discrete real data and the continuous softmax outputs, so the generator may learn to produce extremely "spiky" predictions. Therefore, this training procedure creates a major inconsistency. Specifically, during testing, the generator has to sample a sequence of discrete tokens from the distribution, whereas during training, it is only trained to generate a feasible expectation, which may result in non-realistic texts.

**Score Function-Based Gradient Estimator**  The score function-based estimator [12, 13], also known as REINFORCE [30], is a common solution for addressing the non-differentiable issue mentioned above. By directly parametrizing the probability mass function (PMF) of the sample $x$ as $p_\theta(x)$ and using the identity $\nabla_\theta p_\theta = p_\theta \nabla_\theta \log p_\theta$, the gradient of the expectation of a function $f$ can be written as

$$\nabla_\theta \mathop{\mathbb{E}}_{x \sim p_\theta} [f(x)] = \mathop{\mathbb{E}}_{x \sim p_\theta} [f(x) \nabla_\theta \log p_\theta(x)]. \tag{2}$$

Note that $f$ can be *non-differentiable*. Although REINFORCE is an *unbiased* estimator, it still has a number of disadvantages such as *high variance*, *low sample efficiency* and *credit assignment* problems. Therefore, a lot of efforts have been devoted to reducing the variance including providing per-word rewards [5, 9, 11, 32] and other methods[15, 16].

**Straight-Through Gradient Estimator**  Another heuristic approach is to utilize a straight-through gradient estimator [2]. The basic idea is to treat the discrete operation as if it had been a differentiable proxy during the backward pass. Specifically, using the straight-through estimator with PMF as the proxy of stochastic categorical one-hot vector [18], the gradient of a function $f$ with respect to parameter $\theta$ can be written as

$$\hat{g}_\theta^{\text{ST}} \coloneqq \mathop{\mathbb{E}}_{x \sim p_\theta} [\nabla_{\mathbf{h}} f(\mathbf{h})|_{\mathbf{h}=\text{one-hot}(x)}] \cdot \nabla_\theta \mathbf{p}_\theta, \tag{3}$$

$$\mathbf{p}_\theta \coloneqq [p_\theta(\text{"apple"}), \dots, p_\theta(\text{"zebra"})], \tag{4}$$

where $\mathbf{p}_\theta$ is a vector of probabilities of all categories. The deterministic and dense $\nabla_\theta \mathbf{p}_\theta$ term considering all categories results in a *biased* but *low-variance* estimator [16].

## 3 Proposed Method

In a general setting for probabilistic NLG, a discrete token sequence $\mathbf{x} \in V^T$ is generated through an auto-regressive process, where $V$ is the vocabulary set and $T$ is the length of the sentence. At each step $t$, the generator samples a new token $x_t$ from a soft policy $\pi_\theta$ given the prefix $\mathbf{x}_{<t}$, where the probability of generating a sequence $\mathbf{x}$ is:

$$p_\theta(\mathbf{x}) = \pi_\theta(\mathbf{x}) = \prod_{t=1}^{T} \pi_\theta(x_t \mid \mathbf{x}_{<t}). \tag{5}$$

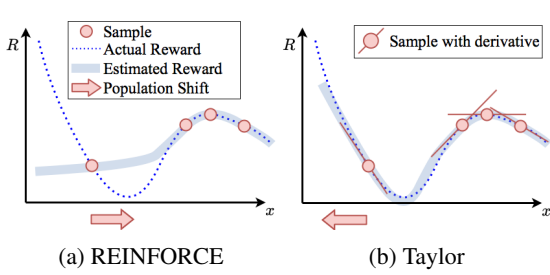

(a) REINFORCE

(b) Taylor

Figure 1: Illustration of policy update estimation.

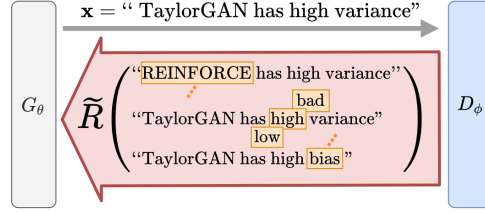

Figure 2: When a sentence $\mathbf{x}$ is fed to the $D_\phi$, rewards of all sentences which differ $\mathbf{x}$ by 1 token are provided through the backward pass.

In the context of GANs with *score function-based approaches*, $\mathbf{x}$ is fed to the discriminator for its reward $R(\mathbf{x})$ after all tokens are generated. The goal of the generator is to maximize the expected reward; therefore $\pi_\theta$ is updated by $\nabla_\theta \mathbb{E}_{\mathbf{x} \sim \pi_\theta} [R(\mathbf{x})]$.

This approach has been adopted to train NLG models with better global consistency than using MLE [9], but is only feasible with a large batch size that reduces the variance of REINFORCE. For a more practical and efficient score function-based method, we propose TaylorGAN, which consists of three key components: **Taylor estimator** for variance reduction, **discriminator constraint** for learning reliable rewards and **entropy maximization** for avoiding mode dropping.

## 3.1 Taylor Estimator

The discriminator is treated as a black box reward provider in the previous score function-based approaches due to non-differentiability of the discrete input space. Ignoring the derivative of $R$, the generator can only learn through the statistics of random guesses. Noisy and inaccurate updates may happen especially when samples are insufficient as demonstrated in Figure 1a. The sensitivity to samples results in *high variance* and *low sample efficiency* problems. Moreover, it is hard to assign proper credits to the values of each generated $x_t$ since $R$ is provided on per-sequence level. Our Taylor estimator mitigates these problems by taking into account neighboring sentences (illustrated in Figure 2), whose rewards can be efficiently approximated with the gradient information from the discriminator.

### 3.1.1 Augmenting Samples by Taylor Expansion

Usually, the reward $R(\mathbf{x})$ of discrete sequence $\mathbf{x}$ is obtained with an embedding operation $E : V^T \to \mathbb{R}^{d_E \times T}$ and a differentiable function $R_E : \mathbb{R}^{d_E \times T} \to \mathbb{R}$ on the embedding space:

$$R(\mathbf{x}) := R_E(E(\mathbf{x})), \tag{6}$$

$$E(\mathbf{x}) := [\mathbf{e}_{x_1}, \dots, \mathbf{e}_{x_T}], \tag{7}$$

where $\mathbf{e}_v \in \mathbb{R}^{d_E}$ is the $d_E$-dimensional column vector representation of the token $v$.

With this particular construction, $R(\mathbf{y})$ can be approximated by the first-order Taylor expansion given $R(\mathbf{x})$ and $\nabla_{\mathbf{E}} R_E(\mathbf{E})$ computed by backpropagation in the embedding layer's output $\mathbf{E} \in \mathbb{R}^{d_E \times T}$, which $\mathbf{y}$ is any sequence $\in V^T$ and $\mathrm{vec}(\cdot)$ denotes the vectorization operation [24] (also known as the flatten operation):

$$\widetilde{R}(\mathbf{x} \to \mathbf{y}) := R(\mathbf{x}) + \mathrm{vec}\left(E(\mathbf{y}) - E(\mathbf{x})\right) \cdot \mathrm{vec}\left(\nabla_{\mathbf{E}} R_E(\mathbf{E})\Big|_{\mathbf{E}=E(\mathbf{x})}\right) \tag{8}$$

$$= R(\mathbf{y}) + \epsilon(\mathbf{x} \to \mathbf{y}),$$

$$|\epsilon(\mathbf{x} \to \mathbf{y})| = \mathcal{O}(\|E(\mathbf{y}) - E(\mathbf{x})\|_2^2). \tag{9}$$

We denote $\widetilde{R}(\mathbf{x} \to \mathbf{y})$ to the first-order approximation of $R(\mathbf{y})$ around $E(\mathbf{x})$, and $\epsilon$ is the first-order Taylor remainder. The benefit of utilizing gradient information is shown in Figure 1b. With $\widetilde{R}$ defined on the tangent plane in the embedding space, the update direction becomes more accurate and less sensitive to the sample $\mathbf{x}$.

However, the approximation is only accurate in the neighborhood of $\mathbf{x}$, because $\epsilon$ increases with the distance from $\mathbf{x}$ in the embedding space. To ensure the neighborhood relation, we can draw $\mathbf{y}$ used in (8) from a joint distribution $\Gamma(\mathbf{x}, \mathbf{y})$, which will be specified in the next subsection, instead of applying the policy $\pi_\theta$. On this off-policy procedure, any expectation on distribution $\pi_\theta$ can still be estimated by importance sampling with likelihood-ratio $I(\mathbf{y}) = \dfrac{\pi_\theta(\mathbf{y})}{\sum_{\mathbf{x}'} \Gamma(\mathbf{x}', \mathbf{y})}$:

$$\mathop{\mathbb{E}}_{\mathbf{y} \sim \pi_\theta} [f(\mathbf{y}; R(\mathbf{y}))] = \mathop{\mathbb{E}}_{(\mathbf{x}, \mathbf{y}) \sim \Gamma} [I(\mathbf{y}) f(\mathbf{y}; R(\mathbf{y}))] \approx \mathop{\mathbb{E}}_{\mathbf{x} \sim \pi_\theta} [\sum_{\mathbf{y}} \frac{\Gamma(\mathbf{x}, \mathbf{y})}{\pi_\theta(\mathbf{x})} I(\mathbf{y}) f(\mathbf{y}; \widetilde{R}(\mathbf{x} \to \mathbf{y}))]. \quad (10)$$

By (10), the reward signal is augmented with neighboring *target* $\mathbf{y}$s, which are approximated from a single *proposal* $\mathbf{x}$; therefore, the estimation is more efficient than previous approaches.

Combining Taylor estimation, off-policy sampling with the REINFORCE estimator in (2) and sampling $\mathbf{y}$ from different joint distribution $\Gamma_t$ of sequence pair $\mathbf{x}, \mathbf{y}$ at each step $t$:

$$\mathop{\mathbb{E}}_{\mathbf{y} \sim \pi_\theta} [R(\mathbf{y}) \nabla_\theta \log \pi_\theta(\mathbf{y})] = \sum_{t=1}^{T} \mathop{\mathbb{E}}_{\mathbf{y} \sim \pi_\theta} [R(\mathbf{y}) \nabla_\theta \log \pi_\theta(y_t \mid \mathbf{y}_{<t})]$$

$$\approx \sum_{t=1}^{T} \mathop{\mathbb{E}}_{\mathbf{x} \sim \pi_\theta} [\sum_{\mathbf{y}} \frac{\Gamma_t(\mathbf{x}, \mathbf{y})}{\pi_\theta(\mathbf{x})} I_t(\mathbf{y}) \widetilde{R}(\mathbf{x} \to \mathbf{y}) \nabla_\theta \log \pi_\theta(y_t \mid \mathbf{y}_{<t})]$$

$$= \mathop{\mathbb{E}}_{\mathbf{x} \sim \pi_\theta} [\sum_{t=1}^{T} \sum_{\mathbf{y}} \frac{\Gamma_t(\mathbf{x}, \mathbf{y})}{\pi_\theta(\mathbf{x})} I_t(\mathbf{y}) \widetilde{R}(\mathbf{x} \to \mathbf{y}) \nabla_\theta \log \pi_\theta(y_t \mid \mathbf{y}_{<t})]. \quad (11)$$

$\Gamma_t(\mathbf{y} \mid \mathbf{x}) := \dfrac{\Gamma_t(\mathbf{x}, \mathbf{y})}{\pi_\theta(\mathbf{x})}$ is defined as the *transition density* that brings signal $\widetilde{R}(\mathbf{x} \to \mathbf{y})$ to reward $\pi_\theta$ for generating $y_t$ given the prefix $\mathbf{y}_{<t}$ when $\mathbf{x}$ is sampled. *Credit assignment* is performed with $\Gamma_t$ and $\nabla_{\mathbf{E}} R_E$ in (8) computed for every step $t$.

### 3.1.2 Hamming Transition

After developing the general concept of the samples augmentation, we need $\Gamma_t$ to establish the neighborhood in the embedding space, and we also enforce $\mathbf{y}_{<t} = \mathbf{x}_{<t}$ to avoid extra computation for $\pi_\theta(\cdot \mid \mathbf{y}_{<t})$ in (11). In this work, we choose the followings:

$$\Gamma_t(\mathbf{y} \mid \mathbf{x}) := \begin{cases} K(y_t \mid x_t) & , \mathbf{x}_{<t} = \mathbf{y}_{<t} \text{ and } \mathbf{x}_{>t} = \mathbf{y}_{>t}, \\ 0 & , \text{otherwise}. \end{cases} \quad (12)$$

$$K(u \mid v) := C(v) \cdot \exp(-\frac{\|\mathbf{e}_u - \mathbf{e}_v\|_2^2}{2\Lambda^2}), \quad (13)$$

where $K$ is a Gaussian kernel function on the embedding space, $C(v)$ is the normalization constant, and $\Lambda > 0$ is defined as the *bandwidth* parameter of the transition.

Since $\mathbf{y}$ only differs from $\mathbf{x}$ by one token, the support of $\Gamma_t$ becomes the projection of *unit Hamming sphere* (UHS) on axis $t$:

$$\text{supp}\,(\Gamma_t(\cdot \mid \mathbf{x})) = \text{UHS}_t(\mathbf{x}) := \{[\mathbf{x}_{<t}, v, \mathbf{x}_{>t}] \mid v \in V\}. \quad (14)$$

Furthermore, all $\widetilde{R}$ can be easily computed on this support (detailed in Appendix A).

To calculate the likelihood-ratio $I_t$, we need $\pi_\theta([\mathbf{x}_{<t}, v, \mathbf{x}_{>t}])$ for arbitrary $v$, which needs much more computation. We therefore ignore the effect of $v$ on the rest of auto-regressive sampling,

$$\frac{\pi_\theta([\mathbf{x}_{<t}, v, \mathbf{x}_{>t}])}{\pi_\theta([\mathbf{x}_{<t}, u, \mathbf{x}_{>t}])} = \frac{\pi_\theta(v \mid \mathbf{x}_{<t})}{\pi_\theta(u \mid \mathbf{x}_{<t})} \frac{\pi_\theta(\mathbf{x}_{>t} \mid \mathbf{x}_{<t}, v)}{\pi_\theta(\mathbf{x}_{>t} \mid \mathbf{x}_{<t}, u)} \approx \frac{\pi_\theta(v \mid \mathbf{x}_{<t})}{\pi_\theta(u \mid \mathbf{x}_{<t})}, \quad (15)$$

then $I_t$ is simplified as:

$$I_t(\mathbf{y}) = \frac{\pi_\theta(\mathbf{y})}{\sum_{\mathbf{x}' \in \text{UHS}_t(\mathbf{y})} \Gamma_t(\mathbf{y} \mid \mathbf{x}') \pi_\theta(\mathbf{x}')} \approx \frac{\pi_\theta(y_t \mid \mathbf{y}_{<t})}{\sum_{u \in V} K(y_t \mid u) \pi_\theta(u \mid \mathbf{y}_{<t})}. \quad (16)$$

The bias introduced by this simplification can be eliminated after replacing the reward with a partial evaluation function independent of $\mathbf{x}_{>t}$ (proved in Appendix B). Due to the additional complexity, we leave this variation for future work.

After subtracting rewards with the bias-free baseline $b$ (proved in Appendix B), the Taylor estimator with Hamming transition is constructed as follows:

$$\hat{g}_\theta^{\text{Taylor}} := \frac{1}{N} \sum_{n=1}^{N} \sum_{t=1}^{T} \sum_{\mathbf{y} \in \text{UHS}_t(\mathbf{x}^{(n)})} \widetilde{A}_t(\mathbf{x}^{(n)} \to \mathbf{y}) \nabla_\theta \log \pi_\theta(y_t \mid \mathbf{x}_{<t}^{(n)}), \tag{17}$$

$$\widetilde{A}_t(\mathbf{x} \to \mathbf{y}) := \frac{K(y_t \mid x_t) \pi_\theta(y_t \mid \mathbf{x}_{<t})}{\sum_{u \in V} K(y_t \mid u) \pi_\theta(u \mid \mathbf{x}_{<t})} (\widetilde{R}(\mathbf{x} \to \mathbf{y}) - b), \tag{18}$$

$$b := \text{Exponential-Moving-Average}(\frac{1}{N} \sum_{n=1}^{N} R(\mathbf{x}^{(n)})), \tag{19}$$

where $N$ is batch size of proposal $\mathbf{x}$s and $\widetilde{A}_t$ is defined as the *advantage* of replacing $x_t$ by $y_t$. The dense form of $\hat{g}_\theta^{\text{Taylor}}$ providing rewards of all sequences on the unit Hamming sphere (illustrated in Figure 2) can also be interpreted as *pseudo exploration*.

**Bias-Variance Trade-Off**   By adjusting bandwidth $\Lambda$ in (13), the Taylor estimator can interpolate between REINFORCE and Straight-Through estimators (proved in Appendix C), which are considered as unbiased and low variance approaches respectively.

- $\Lambda \to 0 \Rightarrow K(u \mid v) \to \mathbb{I}(u = v)$
  No transition or approximation is applied: $\hat{g}_\theta^{\text{Taylor}} \to \hat{g}_\theta^{\text{REINFORCE}}$.
- $\Lambda \to \infty \Rightarrow K(u \mid v) \to 1 / |V|$
  Uniform transition ignoring the neighborhood requirement: $\hat{g}_\theta^{\text{Taylor}} \to \hat{g}_\theta^{\text{ST}}$.

## 3.2   Discriminator Constraint

After constructing the gradient estimator, we choose a proper reward for NLG. We adopt one of the loss functions from Zhou *et al.* [35] for informative gradients:

$$\mathcal{L}_D := - \mathop{\mathbb{E}}_{\mathbf{x} \sim p_{data}} [\log D_\phi(\mathbf{x})] - \mathop{\mathbb{E}}_{\mathbf{x} \sim p_\theta} [\log(1 - D_\phi(\mathbf{x}))] + \mathcal{L}_{\text{reg}}, \tag{20}$$

$$R(\mathbf{x}) := \log \frac{D_\phi(\mathbf{x})}{1 - D_\phi(\mathbf{x})} = \text{sigmoid}^{-1}(D_\phi(\mathbf{x})). \tag{21}$$

This reward is the sigmoid logit, whose unboundedness and absence of the final non-linear sigmoid transformation benefit our Taylor estimator.

We further bound the distance between all pairs of embedding vectors along with Lipschitz constant of $R_E$. Incorporating this constraint to the derivation by Zhou *et al.* [35], the optimal discriminator minimizing (20) can guide the generator towards real samples on the basis of Hamming distance:

$$\|R(\mathbf{x}) - R(\mathbf{y})\|^2 \le \|R_E\|_{Lip}^2 \cdot \sum_{t=1}^{T} \|\mathbf{e}_{x_t} - \mathbf{e}_{y_t}\|_2^2$$

$$\le \|R_E\|_{Lip}^2 \cdot \sup_{u,v \in V} \|\mathbf{e}_u - \mathbf{e}_v\|_2^2 \cdot \sum_{t=1}^{T} \mathbb{I}(x_t \ne y_t)$$

$$= \mathcal{O}(\text{Hamming-distance}(\mathbf{x}, \mathbf{y})). \tag{22}$$

The generator can easily discover better sentences on this metric by with Taylor estimator because according to (14), the augmented rewards come from the unit Hamming sphere.

Lipschitz and embedding constraints are implemented by spectral norm regularization [31] on all layers except the embedding layer and $\ell_2$ penalty over threshold $M$ on word vectors respectively:

$$\mathcal{L}_{\text{reg}} = \frac{\lambda_{SN}}{2} \sum_{l=1}^{L} \sigma(W^l)^2 + \frac{\lambda_E}{2|V|} \sum_{v \in V} \max(\|\mathbf{e}_v\|_2^2 - M^2, 0), \tag{23}$$

where $\sigma(W^l)$ is the largest singular value of $l$th layer's weight $W^l$ and $L$ is the number of layers after the embedding layer in $D_\phi$.

As proven by Arjovsky and Bottou [1], using objectives in (20), (21) without regularization is equivalent to minimizing reverse Kullback–Leibler divergence. This distance measure assigns extremely low cost to the dropped modes. This issue may occurs when $\lambda_{SN}, \lambda_E$ are not high enough. In contrast, high $\lambda_{SN}, \lambda_E$ weaken the discriminator making it far from optimality. As a result, we switch to the alternative in the next subsection to prevent model dropping.

### 3.3 Entropy Maximization

Entropy maximization overcomes the mode dropping issue by encouraging exploration and diversity [10, 23]. In previous work for reinforcement learning, a bonus term is directly added to reward such as $(R - \lambda_{\mathcal{H}} \log \pi_\theta) \nabla_\theta \log \pi_\theta$. This term introduces extra randomness to the estimator and becomes a source of variance. Whence, we choose an alternative form considering all possible token (detailed in Appendix D) and obtain a dense objective as proposed Taylor estimator:

$$\nabla_\theta \mathcal{H}(\pi_\theta(\cdot \mid \mathbf{x}_{<t})) = -\nabla_\theta \sum_{v \in V} \pi_\theta(v \mid \mathbf{x}_{<t}) \log \pi_\theta(v \mid \mathbf{x}_{<t}). \tag{24}$$

Adding the above term to (17), the update-formula of the generator's variable $\theta$ becomes:

$$\delta\theta = \hat{g}_\theta^{\text{Taylor}} + \frac{\lambda_{\mathcal{H}}}{N} \sum_{n=1}^{N} \sum_{t=1}^{T} \nabla_\theta \mathcal{H}(\pi_\theta(\cdot \mid \mathbf{x}_{<t}^{(n)})). \tag{25}$$

With the proposed components, our TaylorGAN is capable of training NLG models efficiently without a large batch size and maximum likelihood pre-training.

## 4 Experiments

In order to evaluate the effectiveness of the proposed approach, we perform a set of NLG experiments and analyze the results.

### 4.1 Experimental Setup

The dataset used in the experiments is **EMNLP 2017 News**, where sentences have a maximum length of 50 tokens and a vocabulary of 5.3k words after performing lowercase. Training and validation data consist of 269k and 10k sentences respectively.[2] Additional results on COCO image caption dataset can be found in Appendix G.

All models (detailed in Appendix E) are trained for at most 200 epochs, namely 800k training steps with the batch size $N = 64$. We save the model every epoch, and select the one with the best validation FED. Each training takes approximately 1 day on Nvidia Tesla V100 GPU.

Following Caccia *et al.* [4], we apply temperature control by adjusting the softmax temperature parameter at test time in order to balance the trade-off between quality and diversity. Increasing the temperature makes probabilities more uniform, which leads to diverse but low-quality samples while reducing it leads to high-quality yet less diverse samples.

### 4.2 Evaluation Metrics

We evaluate our model's ability to generate realistic and diverse texts with n-gram based metric, Fréchet Embedding Distance and language model based metric.

**N-gram Based** BLEU [26] and Self-BLEU [36] capture text quality and diversity respectively. Smoothing 1 proposed in [6] is used with $\epsilon = 0.1$ for both (detailed in Appendix F).

- BLEU: a modified form of n-gram precision, measures local consistency between a set of reference and a set of candidate texts.

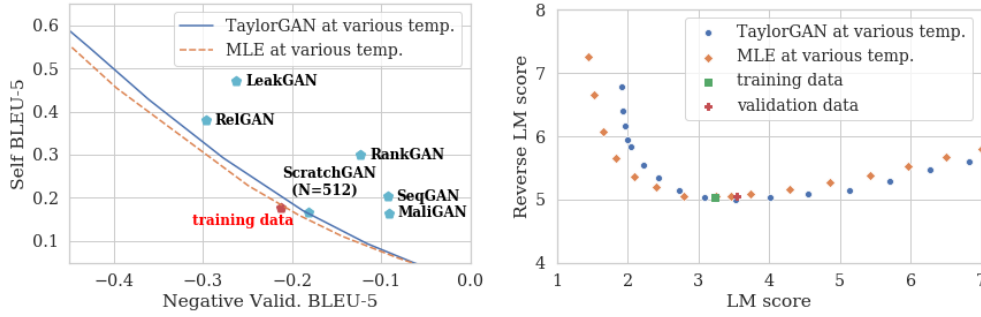

(a) Negative BLEU-5 versus Self-BLEU-5.　　(b) Language- and reverse language-model scores.

Figure 3: Temperature sweeps of BLEU (left) and language model scores on (right) on EMNLP 2017 News. Left and lower is better.

- Self-BLEU: A version of BLEU in which both the reference and candidate texts are drawn from the same corpus. A high Self-BLEU means that members within the corpus are too highly correlated, indicating a lack of diversity. The size of each set is always 5k in this work.

**Fréchet Embedding Distance**　　Fréchet Embedding Distance (FED) [9] uses an Universal Sentence Encoder[3] to measure sentimental similarity, global consistency and diversity between reference and candidate texts. FED is always computed with 10k against 10k samples in this work.

**Language Model Scoring**　　Following Caccia *et al.* [4], Zhao *et al.* [34], we also evaluate the quality and the diversity of generated samples by training external language model (LM) and reverse language model (RLM). We use the architecture in de Masson dAutume *et al.* [9] for both.

- LM: We use a language model trained on real data to estimate the *negative log likelihood per word* of generated sentences. Samples which the language model considers improper, such as ungrammatical or nonsense sentences, have a low score.
- RLM: If we instead train a language model on generated data while evaluating on real data, we can judge the diversity of the generated samples. Low-diversity generated data leads to an overfitted model that generalizes poorly on real data, as indicated by a low score. Models are always trained with 10k samples in this work.

**Perplexity**　　Apart from using external language model, we can also use our generator, since it is also an autoregressive language model that produces the probability of given sequence as shown in (5). We can evaluate the generator's perplexity, which is the inverse of the per-word probability, for generating given real data. If there is a mode dropping problem, it will result in high perplexity.

## 5　Results & Discussion

In this section, we compare our results with previous approaches including MLE, LeakGAN [17], MaliGAN [5], RankGAN [22], RelGAN [25], ScratchGAN [9] and SeqGAN [32] [4] from different perspectives via the evaluation metrics presented in the last section. Furthermore, a quantitative study is performed to know the contribution of each technique. Samples from MLE, ScratchGAN, and TaylorGAN can also be seen in Appendix H, alongside with training data.

### 5.1　Quality and Diversity

On metrics of local text consistency and diversity, TaylorGAN significantly outperforms prior GAN-based models which rely heavily on pre-training except ScratchGAN. The BLEU/Self-BLEU

Table 1: FED & LM scores on EMNLP 2017 News. Lower is better. Batch size $N = 64$ if not specified. † follows the settings of [9] and ‡ are pre-trained by MLE.

| Model | FED | | LM |
|---|---|---|---|
| | **Train** | **Val.** | **Score** |
| Training data | 0.0050 | 0.0120 | 3.22 |
| MLE† | 0.0100 | 0.0194 | **3.43** |
| SeqGAN‡ | 0.1234 | 0.1422 | 6.09 |
| MaliGAN‡ | 0.1280 | 0.1504 | 6.30 |
| RankGAN‡ | 0.1418 | 0.1431 | 5.76 |
| LeakGAN‡ | 0.0718 | 0.0691 | 4.90 |
| RelGAN‡ | 0.0462 | 0.0408 | 3.49 |
| ScratchGAN | 0.0301 | 0.0390 | 4,96 |
| ScratchGAN ($N = 512$) | 0.0153 | 0.0194 | 4.46 |
| TaylorGAN | 0.0105 | **0.0149** | 4.02 |

Table 2: Quantitative study on EMNLP 2017 News validation data. Lower is better.

| $\Lambda$ | $\lambda_{SN}$ | $\lambda_{\mathcal{H}}$ | FED | Perplexity |
|---|---|---|---|---|
| Gumbel-Softmax [18] | | | | |
| N/A | 0.07 | 0.02 | 0.0218 | 6369 |
| REINFORCE ($\Lambda \to 0$) | | | | |
| | 0.07 | 0.02 | 0.0181 | 72 |
| | 0.10 | 0.02 | 0.0183 | 64 |
| | 0.03 | 0.02 | 0.0221 | 91 |
| | 0.01 | 0.02 | 0.0410 | 133 |
| | 0.07 | 0.00 | 0.0203 | 211 |
| Straight-Through ($\Lambda \to \infty$) | | | | |
| | 0.07 | 0.02 | 0.0449 | 116 |
| Taylor ($\Lambda = 0.5$ as default) | | | | |
| 0.50 | 0.07 | 0.02 | **0.0149** | 72 |
| 0.25 | 0.07 | 0.02 | 0.0179 | 75 |
| 1.00 | 0.07 | 0.02 | 0.0199 | **60** |
| 0.50 | 0.10 | 0.02 | 0.0178 | 64 |
| 0.50 | 0.03 | 0.02 | 0.0183 | 91 |
| 0.50 | 0.01 | 0.02 | 0.0221 | 99 |
| 0.50 | 0.07 | 0.00 | 0.0200 | 310 |

temperature sweeps shown in Figure 3a for TaylorGAN indicate that TaylorGAN approaches the performance of MLE model. On the other hand, in Figure 3b, the LM scores show similar results for TaylorGAN and MLE model.

## 5.2 Global Consistency

TaylorGAN improves global consistency comparing to prior works and language model on LM score and FED score, respectively. As shown in Table 1, our method reduces the gap of LM score between GAN-based method and MLE, where the latter is directly trained to minimize the negative log likelihood. Besides, in Table 1 we show that TaylorGAN outperforms MLE and other MLE pre-trained or large batch GAN-based methods on FED score.

## 5.3 Quantitative Study

We show the influence of gradient estimator, $\Lambda$, $\lambda_{SN}$, and $\lambda_{\mathcal{H}}$ in Table 2 on validation performance, and observe that:

- The Taylor estimator outperforms Gumbel-Softmax, REINFORCE and Straight-Through baselines on FED.
- We argue that the inferior performance of Gumbel-Softmax is the consequence of biased and spiky distribution explained in Section 2 and the unusually high perplexity on real data, even with temperature annealing during the training phase [18].
- High $\Lambda$ results in better perplexity. We argue that it is due to the generator being forced to explore by the Taylor estimator in this case.
- The Taylor estimator is less reliant on discriminator constraint, showing its potential of transfer learning using a reward network trained from another source.
- Low $\lambda_{SN}$ leads to worse perplexity while high $\lambda_{SN}$ leads to worse FED. It supports the statement we make in Section 3.2.
- Entropy maximization significantly reduces mode dropping indicated by much lower perplexity compared to $\lambda_{\mathcal{H}} = 0$.

## 6    Conclusion and Future Work

In this work, we have presented TaylorGAN, an effective method for training an NLG model without MLE pre-training. Starting from REINFORCE, we derive a new estimator with a proper reward function that improves the sample efficiency and reduces the variance. The benchmark experiments demonstrate that the proposed TaylorGAN improves the quality/diversity of generated texts as measured by various standard metrics. In the future, we plan to improve our estimator for achieving low bias together with low variance, and apply it to other fields such as model-based reinforcement learning.

## Broader Impact

Improving text generation may have a wide range of beneficial impact across many domains. This includes human-machine collaboration of code, literature or even music, chatbots, question-answering systems, etc. However, this technology may be misused whether deliberately or not.

Because our model learns the underlying distribution of datasets like any other language model, it inherently risks producing biased or offensive content reflective of the training data. Studies have shown that language models could produce biased content with respect to gender, race, religion, while modeling texts from the web [29].

Better text generation could lower costs of disinformation campaigns and even weaken our detection ability of synthetic texts. Studies found that extremist groups can use language models for misuse specifically by finetuning the model on corresponding ideological positions [29].

## Acknowledgments and Disclosure of Funding

We would like to thank all reviewers for their insightful comments; Alvin Chiang, Chien-Fu Lin, Chi-Liang Liu, Po-Hsien Chu, Yi-En Tsai and Chung-Yang (Ric) Huang for thoughtful discussions; Shu-Wen Yang and Huan Lin for technical supports. We are grateful to the National Center for High-performance Computing for computing resources. This work was financially supported from the Young Scholar Fellowship Program by Ministry of Science and Technology (MOST) in Taiwan, under Grant 109-2636-E-002-026 and Grant 109-2636-E-002-027.

## Footnotes

[1]The source code and data are available at https://github.com/MiuLab/TaylorGAN/

[2]The data is at https://github.com/pclucas14/GansFallingShort/tree/master/real_data_experiments/data/news

[3]https://tfhub.dev/google/universal-sentence-encoder/2

[4]Reproduced by running the released code on https://github.com/geek-ai/Texygen/ and https://github.com/deepmind/deepmind-research/tree/master/scratchgan

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
