[Supplementary Material]

# Appendix A   Implementation of Taylor Expansion on Unit Hamming Sphere

Following Section 3.1.2, since all **y**s are drawn from the unit Hamming sphere centered at **x**, (8) becomes a special case:

$$
\begin{aligned}
\widetilde{R}(\mathbf{x} \to [\mathbf{x}_{<t}, v, \mathbf{x}_{>t}]) &= R(\mathbf{x}) + (\mathbf{e}_v - \mathbf{e}_{x_t}) \cdot \nabla_{\mathbf{E}_t} R_E(\mathbf{E}) \Big|_{\mathbf{E}=E(\mathbf{x})} \\
&= R(\mathbf{x}) + \mathbf{e}_v \cdot \nabla_{\mathbf{E}_t} R_E(\mathbf{E}) - \mathbf{e}_{x_t} \cdot \nabla_{\mathbf{E}_t} R_E(\mathbf{E}) \Big|_{\mathbf{E}=E(\mathbf{x})} \\
&= R(\mathbf{x}) + \mathbf{e}_v^\top \nabla_{\mathbf{E}_t} R_E(\mathbf{E}) - \mathbf{1}_{d_E}^\top (\mathbf{e}_{x_t} \odot \nabla_{\mathbf{E}_t} R_E(\mathbf{E})) \Big|_{\mathbf{E}=E(\mathbf{x})},
\end{aligned}
\tag{26}
$$

where $\mathbf{E}_t \in \mathbb{R}^{d_E}$ is the $t$th column of $\mathbf{E}$, $\mathbf{1}$ is matrix of ones, $\top$ stands for transpose, and $\odot$ stands for element-wise multiplication.

$\big\{\widetilde{R}(\mathbf{x} \to \mathbf{y}) \mid \mathbf{y} \in \bigcup_{t=1}^T \mathrm{UHS}_t(\mathbf{x})\big\}$ can be computed efficiently by stacking the results of (26) to a matrix $\widetilde{\mathbf{R}} \in \mathbb{R}^{|V| \times T}$ (we simplify $\nabla_{\mathbf{E}} R_E(\mathbf{E})\big|_{\mathbf{E}=E(\mathbf{x})}$ as $\delta \mathbf{E}$ for convenience):

$$
\begin{aligned}
\widetilde{\mathbf{R}}(\mathbf{x}) &:= \begin{bmatrix} \widetilde{R}(\mathbf{x} \to [\text{``apple''}, \mathbf{x}_{>1}]) & \dots & \widetilde{R}(\mathbf{x} \to [\mathbf{x}_{<T}, \text{``apple''}]) \\ \vdots & \ddots & \vdots \\ \widetilde{R}(\mathbf{x} \to [\text{``zebra''}, \mathbf{x}_{>1}]) & \dots & \widetilde{R}(\mathbf{x} \to [\mathbf{x}_{<T}, \text{``zebra''}]) \end{bmatrix} \\
&= R(\mathbf{x}) \mathbf{1}_{|V| \times T} + W_E\, \delta \mathbf{E} - \mathbf{1}_{|V| \times d_E} (E(\mathbf{x}) \odot \delta \mathbf{E}),
\end{aligned}
\tag{27}
$$

where $W_E := \begin{bmatrix} \mathbf{e}_{\text{``apple''}}^\top \\ \vdots \\ \mathbf{e}_{\text{``zebra''}}^\top \end{bmatrix}$ is the $|V| \times d_E$ embedding matrix of all tokens and $\delta \mathbf{E}$ can be computed by backpropagation.

With the above operations, there is no need to explicitly collect the neighboring sequences on unit Hamming spheres.

# Appendix B   Properties of Suffix Probability Simplification

Following Section 3.1.2 and replacing $\widetilde{R}$ by a function $f$ independent to suffix $\mathbf{x}_{>t}$, the expectation of the Taylor estimator becomes

$$
\begin{aligned}
&\mathbb{E}_{\mathbf{x} \sim \pi_\theta} \Big[ \sum_{\mathbf{y} \in \mathrm{UHS}_t(\mathbf{x})} \frac{K(y_t \mid x_t)\pi_\theta(y_t \mid \mathbf{x}_{<t})}{\sum_{u \in V} K(v \mid u)\pi_\theta(u \mid \mathbf{x}_{<t})} f(\mathbf{x}_{<t}, v) \nabla_\theta \log \pi_\theta(v \mid \mathbf{x}_{<t}) \Big] \\
&= \mathbb{E}_{\mathbf{x}_{\le t} \sim \pi_\theta} \Big[ \sum_{v \in V} \frac{K(v \mid x_t)\pi_\theta(v \mid \mathbf{x}_{<t})}{\sum_{u \in V} K(v \mid u)\pi_\theta(u \mid \mathbf{x}_{<t})} f(\mathbf{x}_{<t}, v) \nabla_\theta \log \pi_\theta(v \mid \mathbf{x}_{<t}) \Big] \\
&= \mathbb{E}_{\mathbf{x}_{<t} \sim \pi_\theta} \Big[ \sum_{x_t \in V} \pi_\theta(x_t \mid \mathbf{x}_{<t}) \sum_{v \in V} \frac{K(v \mid x_t)\pi_\theta(v \mid \mathbf{x}_{<t})}{\sum_{u \in V} K(v \mid u)\pi_\theta(u \mid \mathbf{x}_{<t})} f(\mathbf{x}_{<t}, v) \nabla_\theta \log \pi_\theta(v \mid \mathbf{x}_{<t}) \Big] \\
&= \mathbb{E}_{\mathbf{x}_{<t} \sim \pi_\theta} \Big[ \sum_{v \in V} \frac{\sum_{x_t \in V} K(v \mid x_t)\pi_\theta(x_t \mid \mathbf{x}_{<t})}{\sum_{u \in V} K(v \mid u)\pi_\theta(u \mid \mathbf{x}_{<t})} f(\mathbf{x}_{<t}, v) \nabla_\theta \pi_\theta(v \mid \mathbf{x}_{<t}) \Big] \\
&= \mathbb{E}_{\mathbf{x}_{<t} \sim \pi_\theta} \Big[ \sum_{v \in V} f(\mathbf{x}_{<t}, v) \nabla_\theta \pi_\theta(v \mid \mathbf{x}_{<t}) \Big] = \nabla_\theta \mathbb{E}_{\mathbf{x}_{\le t} \sim \pi_\theta} [f(\mathbf{x}_{<t}, x_t)].
\end{aligned}
\tag{28}
$$

Let $f(\mathbf{x}_{<t}, v)$ be a constant baseline $b$, we proved that replacing $\widetilde{R}$ with $\widetilde{R} - b$ introduces no bias due to the fact $\nabla_\theta \mathbb{E}[b] = \nabla_\theta b = 0$.

Let $f(\mathbf{x}_{<t}, v)$ be the state-action value function $Q_\theta^\pi(\mathbf{x}_{<t}, v) := \mathbb{E}_{\mathbf{x}_{>t} \sim \pi_\theta(\cdot | \mathbf{x}_{<t}, v)} \big[ R([\mathbf{x}_{<t}, v, \mathbf{x}_{>t}]) \big]$, we proved that the simplified form is an unbiased estimator of $\nabla_\theta \mathbb{E}_{\pi_\theta}[Q]$.

# Appendix C  Equivalence to REINFORCE and Straight-Through Estimators

Follow the discussion in Section 3.1.2 about the special cases of bandwidth $\Lambda$. Due to the property proved in Appendix B, baseline $b$ in (17) is eliminated in the following proofs.

**REINFORCE**  $\Lambda \to 0 \Rightarrow K(u \mid v) \to \mathbb{I}(u = v)$.
By using the identity $\widetilde{R}(\mathbf{x} \to \mathbf{x}) = R(\mathbf{x})$, the Taylor estimator in (17) with respect to a single sample $\mathbf{x}$ becomes

$$
\begin{aligned}
&\sum_{t=1}^{T} \sum_{\mathbf{y} \in \mathrm{UHS}_t(\mathbf{x})} \widetilde{A}_t(\mathbf{x} \to \mathbf{y}) \nabla_\theta \log \pi_\theta(y_t \mid \mathbf{x}_{<t}) \\
=&\sum_{t=1}^{T} \sum_{\mathbf{y} \in \mathrm{UHS}_t(\mathbf{x})} \frac{\mathbb{I}(y_t = x_t) \pi_\theta(y_t \mid \mathbf{x}_{<t})}{\sum_{u \in V} \mathbb{I}(y_t = u) \pi_\theta(u \mid \mathbf{x}_{<t})} \widetilde{R}(\mathbf{x} \to \mathbf{y}) \nabla_\theta \log \pi_\theta(y_t \mid \mathbf{x}_{<t}) \\
=&\sum_{t=1}^{T} \frac{\pi_\theta(x_t \mid \mathbf{x}_{<t})}{\pi_\theta(x_t \mid \mathbf{x}_{<t})} \widetilde{R}(\mathbf{x} \to \mathbf{x}) \nabla_\theta \log \pi_\theta(x_t \mid \mathbf{x}_{<t}) \\
=&\sum_{t=1}^{T} R(\mathbf{x}) \nabla_\theta \log \pi_\theta(x_t \mid \mathbf{x}_{<t}) = R(\mathbf{x}) \nabla_\theta \log \pi_\theta(\mathbf{x}) = \hat{g}_\theta^{\mathrm{REINFORCE}}(\mathbf{x}),
\end{aligned}
\tag{29}
$$

where $\hat{g}^{\mathrm{REINFORCE}}$ is exact the same form as (2).

**Straight-Through Estimator**  $\Lambda \to \infty \Rightarrow K(u \mid v) \to \text{constant}$.
Following (3) and setting $f(\mathbf{h}) = g(W_E^\top \mathbf{h})$, where $\mathbf{h}$ is any $|V|$-dimensional column vector and $g : \mathbb{R}^{d_E} \to \mathbb{R}$, we have

$$
\left. \nabla_\mathbf{h} f(\mathbf{h}) \right|_{\mathbf{h}=\text{one-hot}(x_t)} = \left. \nabla_\mathbf{h}\, g(W_E^\top \mathbf{h}) \right|_{\mathbf{h}=\text{one-hot}(x_t)} = \left. W_E \nabla_\mathbf{e} g(\mathbf{e}) \right|_{\mathbf{e}=W_E^\top \text{one-hot}(x)=\mathbf{e}_x}
$$

$$
\Rightarrow \left. \nabla_\mathbf{h} f(\mathbf{h}) \right|_{\mathbf{h}=\text{one-hot}(x_t)} \cdot \nabla_\theta \mathbf{p}_\theta = \left. \nabla_\mathbf{e} g(\mathbf{e}) \right|_{\mathbf{e}=\mathbf{e}_x} \cdot \sum_{v \in V} \mathbf{e}_v \nabla_\theta p_\theta(v),
\tag{30}
$$

which implies the equivalence between (3) and using the expectation of word vector as the proxy of stochastic embedding operation.

Using the above property on $\pi_\theta(\cdot \mid \mathbf{x}_{<t})$ instead of $p_\theta$, identity $\sum_{v \in V} \nabla_\theta \pi_\theta(v \mid \mathbf{x}_{<t}) = \nabla_\theta 1 = 0$ and simplifying $\left. \nabla_{\mathbf{E}_t} R_E(\mathbf{E}) \right|_{\mathbf{E}=E(\mathbf{x})}$ as $\delta \mathbf{E}_t$ for convenience, (17) with respect to a single sample $\mathbf{x}$ and step $t$ becomes

$$
\begin{aligned}
&\sum_{\mathbf{y} \in \mathrm{UHS}_t(\mathbf{x})} \widetilde{A}_t(\mathbf{x} \to \mathbf{y}) \nabla_\theta \log \pi_\theta(y_t \mid \mathbf{x}_{<t}) \\
=&\sum_{\mathbf{y} \in \mathrm{UHS}_t(\mathbf{x})} \frac{\pi_\theta(y_t \mid \mathbf{x}_{<t})}{\sum_{u \in V} \pi_\theta(u \mid \mathbf{x}_{<t})} \widetilde{R}(\mathbf{x} \to \mathbf{y}) \nabla_\theta \log \pi_\theta(y_t \mid \mathbf{x}_{<t}) \\
=&\sum_{\mathbf{y} \in \mathrm{UHS}_t(\mathbf{x})} \widetilde{R}(\mathbf{x} \to \mathbf{y}) \nabla_\theta \pi_\theta(y_t \mid \mathbf{x}_{<t}) \\
=&\sum_{v \in V} (R(\mathbf{x}) + (\mathbf{e}_v - \mathbf{e}_{x_t}) \cdot \delta \mathbf{E}_t) \nabla_\theta \pi_\theta(v \mid \mathbf{x}_{<t}) \\
=&\,(R(\mathbf{x}) - \mathbf{e}_{x_t} \cdot \delta \mathbf{E}_t) \underbrace{\sum_{v \in V} \nabla_\theta \pi_\theta(v \mid \mathbf{x}_{<t})}_{0} + \sum_{v \in V} \mathbf{e}_v \cdot \delta \mathbf{E}_t \nabla_\theta \pi_\theta(v \mid \mathbf{x}_{<t}) \\
=&\,\delta \mathbf{E}_t \cdot \sum_{v \in V} \mathbf{e}_v \nabla_\theta \pi_\theta(v \mid \mathbf{x}_{<t}) = \hat{g}_{\theta,t}^{\mathrm{ST}}(\mathbf{x}).
\end{aligned}
\tag{31}
$$

In this case, (17) is equivalent to using (3) on the sampling operation of every step $t$.

# Appendix D  Gradient Estimator of Entropy

In this section, we continue the discussion in Section 3.3 and obtain the form used in (25). The Shannon entropy of discrete samples randomly generated by policy $\pi_\theta$ is defined as

$$\mathcal{H}(\pi) := -\sum_{\mathbf{x}\in V^T} \pi_\theta(\mathbf{x})\log \pi_\theta(\mathbf{x}). \tag{32}$$

To compute the gradient with respect to parameter $\theta$, we apply the same identity $\nabla_\theta \pi_\theta = \pi_\theta \nabla_\theta \log \pi_\theta$ used to derive (2) and get

$$
\begin{aligned}
\nabla_\theta \mathcal{H}(\pi_\theta) &= -\nabla_\theta \sum_{\mathbf{x}\in V^T} \pi_\theta(\mathbf{x})\log \pi_\theta(\mathbf{x}) \\
&= -\sum_{\mathbf{x}\in V^T} \log \pi_\theta(\mathbf{x})\nabla_\theta \pi_\theta(\mathbf{x}) + \pi_\theta(\mathbf{x})\frac{\nabla_\theta \pi_\theta(\mathbf{x})}{\pi_\theta(\mathbf{x})} \\
&= -\sum_{\mathbf{x}\in V^T} \log \pi_\theta(\mathbf{x})\nabla_\theta \pi_\theta(\mathbf{x}) - \underbrace{\sum_{\mathbf{x}\in V^T} \nabla_\theta \pi_\theta(\mathbf{x})}_{0} \\
&= \mathop{\mathbb{E}}_{\mathbf{x}\sim\pi_\theta} [-\log \pi_\theta(\mathbf{x})\nabla_\theta \log \pi_\theta(\mathbf{x})],
\end{aligned}
\tag{33}
$$

which has a similar form as REINFORCE with $R_\mathcal{H}(\mathbf{x}) := -\log \pi_\theta(\mathbf{x})$ as objective. Thus, $R_\mathcal{H}$ can be added to the original reward to encourage diversity.

Furthermore, due to (5), $R_\mathcal{H}(\mathbf{x})$ can be written as $\sum_{t=1}^T r_t(\mathbf{x})$ with $r_t(\mathbf{x}) := -\log \pi_\theta(x_t \mid \mathbf{x}_{<t})$, which provides per-word rewards. By applying causality and discounting the future reward with a factor $\gamma$, we rewrite the update signal of $\theta$ from the entropy bonus as

$$\hat{g}_\theta^{\mathcal{H},\gamma} := \mathop{\mathbb{E}}_{\mathbf{x}\sim\pi_\theta} \Big[\sum_{t=1}^T \nabla_\theta \log \pi_\theta(x_t \mid \mathbf{x}_{<t}) \sum_{t'=t}^T \gamma^{t'-t} r_{t'}(\mathbf{x})\Big]. \tag{34}$$

In this work, we use $\gamma \to 0$ for a greedy reward, which ignores the suffix $\mathbf{x}_{>t}$ and allows us to write a dense form by expanding all possible next token $x_t$:

$$
\begin{aligned}
\hat{g}_\theta^{\mathcal{H},0} &= \mathop{\mathbb{E}}_{\mathbf{x}\sim\pi_\theta} \Big[\sum_{t=1}^T \nabla_\theta \log \pi_\theta(x_t \mid \mathbf{x}_{<t}) \, r_t(\mathbf{x})\Big] \\
&= -\mathop{\mathbb{E}}_{\mathbf{x}\sim\pi_\theta} \Big[\sum_{t=1}^T \nabla_\theta \log \pi_\theta(x_t \mid \mathbf{x}_{<t}) \log \pi_\theta(x_t \mid \mathbf{x}_{<t}))\Big] \\
&= -\sum_{t=1}^T \mathop{\mathbb{E}}_{\mathbf{x}_{<t}\sim\pi_\theta} \Big[\sum_{v\in V} \pi_\theta(v \mid \mathbf{x}_{<t})\nabla_\theta \log \pi_\theta(v \mid \mathbf{x}_{<t}) \log \pi_\theta(v \mid \mathbf{x}_{<t}))\Big] \\
&= -\sum_{t=1}^T \mathop{\mathbb{E}}_{\mathbf{x}_{<t}\sim\pi_\theta} \Big[\sum_{v\in V} \nabla_\theta \pi_\theta(v \mid \mathbf{x}_{<t}) \log \pi_\theta(v \mid \mathbf{x}_{<t}))\Big] \\
&= -\mathop{\mathbb{E}}_{\mathbf{x}\sim\pi_\theta} \Big[\sum_{t=1}^T \sum_{v\in V} \nabla_\theta \pi_\theta(v \mid \mathbf{x}_{<t}) \log \pi_\theta(v \mid \mathbf{x}_{<t}))\Big] = \mathop{\mathbb{E}}_{\mathbf{x}\sim\pi_\theta} \Big[\sum_{t=1}^T \nabla_\theta \mathcal{H}(\pi_\theta(\cdot \mid \mathbf{x}_{<t}))\Big],
\end{aligned}
\tag{35}
$$

where $\mathcal{H}(\pi_\theta(\cdot \mid \mathbf{x}_{<t}))$ is the entropy of next-token distribution. This biased but dense form works well in practice.

# Appendix E  Model Architecture

## E.1  Discriminator

The architecture of the discriminator is shown in Table 3. Exponential Linear Units [8], whose continuous 1$^{\text{st}}$ derivative and sparse 2$^{\text{nd}}$ derivative satisfy the condition of Taylor's theorem in (8), and reduce the norm of Hessian matrix, is used as all non-linear transformations.

| | | |
|---|---|---|
| Embedding | | |
| Conv3-512 ELU | | |
| Conv4-512 ELU | | |
| Mean Pool2 | | |
| Conv3-1024 ELU | | |
| Conv4-1024 ELU | | |
| Global Mean Pool | | |
| Dense 1024 ELU | | |
| Dense 1 Sigmoid | | |

Table 3: Discriminator.

Figure 4: Masking mechanism.

Figure 5: Generator.

To perform consistent convolution on sentences of different lengths, we also implement a special masking mechanism. For all convolution and pooling layers, we mask out input features where the receptive field is out-of-bounds with respect to the unpadded input sentence. As shown in Figure 4, the same padding is performed as if $T_{\text{input}}$ had been 3, even though it is actually 5 (not including the leftmost $\mathbf{0}$ added by original implementation of same padding) with 2 PAD tokens .

### E.2 Generator

The generator we use is an autoregressive probabilistic model with GRU [7] shown in Figure 5. In the linear layer, the output of GRU is projected to the dimension of word vectors and then multiplied by the transpose of embedding matrix [27] to obtain the softmax logits. The start-of-sentence token is fed to the embedding layer at $t = 1$. The vocabulary also contains an end-of-sentence token. If the generator outputs this token at any time step $t_{\text{EOS}}$, the sentence ends and its length is set to $t_{\text{EOS}}$.

## Appendix F  Training Details

The discriminator and generator are updated with a 1:1 ratio and both trained with Adam [20] of learning rate $= 10^{-4}, \beta_1 = 0.5$ and $\beta_2 = 0.999$. Gradients are clipped by maximum global norm $= 10$. $\Lambda = 0.5, \lambda_{SN} = 0.07, \lambda_E = 0.2, M = 1$ and $\lambda_{\mathcal{H}} = 0.02$ if not specified. Baseline $b$ is assigned as the exponential moving average of $R$ with decay rate $= 0.9$. All word vectors are initialized with pre-trained fastText embeddings [3].

**BLEU smoothing**  One of the issue with BLEU is that in the case that a higher order n-gram precision of a sentence is 0, then the BLEU score will be 0, resulting in severely underestimation. This is due to the fact that BLEU is calculated by the geometric mean of precision. To solve this, we replaced the precision by the smoothed version as follows:

$$\text{smoothed-precision}_n := \max \left( \text{precision}_n, \frac{\epsilon}{\text{Count(n-grams)}} \right). \tag{36}$$

We've picked the smoothing factor $\epsilon = 0.1$ as proposed by Chen and Cherry [6].

## Appendix G  COCO Results

Sentences in the COCO dataset have a maximum length of 24 tokens and a vocabulary of 4.6k words after performing lowercase. Training and validation data both consist of 10k sentences.[5] The BLEU/Self-BLEU temperature sweeps is shown in Figure 6.

Figure 6: Temperature sweeps on COCO. Left and lower is better.

Figure 7: Sentence length statistics on EMNLP2017 News.

## Appendix H    Generated Samples

Samples of EMNLP2017 News from MLE, ScratchGAN, and TaylorGAN can be seen in Table 4, alongside with training data. The senetence length statistics is shown in Figure 7.

Samples of COCO from MLE, LeakGAN, MaliGAN, RankGAN, RelGAN, SeqGAN, TextGAN, and TaylorGAN can be seen in Table 5, alongside with training data.

Table 4: Training data, MLE, ScratchGAN, TaylorGAN samples of EMNLP 2017 News with temperature = 1.0. $N$ is the batch size used to train each model.

**Training Data**

- the u . s . has struggled to find an effective ground force to take on isis in syria , where president obama has ruled out a u . s . ground combat role .
- it is a happy and kind world that we live in on this show and that is where i hope we can live in real life .
- while men and women may end up earning roughly the same amount in the same jobs , men are more likely to end up in higher - paying roles in the tech industry .
- but while they were beaten by a better side , the tie did reveal what i think has been city ' s biggest problem this season : they have lost the ability to score against good teams .
- according to facebook ' s policies , accounts can be suspended if law enforcement believe individuals are at risk of harm .

**MLE** ($N = 64$)

- a one - game event and a limited number this dropped by 38 per cent to find out what we could want , to be as much done as we finish as the games .
- the uk government got on the deal when it came to other eu countries that allowed workplace checks for 26 .
- so women i didn ' t want my parents to work hard with me at the time of the task of making it presents .
- black voters feel there may not be enough choice between former president george w . bush and i have to go along with problems and support the people that are in a swing state for president .
- let ' s catch him down his back phone one morning john f kennedy , where he is in a us meeting group with supporters of love not being successful in the fields .

**ScratchGAN** ($N = 512$)

- they were together to be twice a week of a junior doctor who work pushed their home but they can necessarily have to have a size to go one of strikes on .
- democrats cannot overcome a support of mr . barack obama , and obama has an influence we have other aspects of the most powerful president and fought supporting the gop nominee .
- the researchers thought that women looked older for more than boys having been used in terms of the sexual assault from the swedish victims .
- " since i get engaged with the woman you share all wonderful and got out of my foot traveling to the main daily network , everything told me he their sister were dead and showing what sort of position there and , " he said .

**TaylorGAN** ($N = 64$)

- " well , i don ' t think it will ever be a good opportunity , " murray told the press .
- " however , they do not know how much the information needed to be discovered , " the company said in a statement .
- i ' m so much younger than i wanted to , and it didn ' t sense so much for me .
- " i think it was good enough that she was ready to take the time and just let her run this , " she said .
- when you see people coming together , that ' s pretty different from what we ' ve taught in america .

Table 5: Training data, MLE, LeakGAN, MaliGAN, RankGAN, RelGAN, SeqGAN, TaylorGAN samples of
COCO image caption with temperature = 1.0.

**Training Data**

- an industrial kitchen with white walls and stainless steel counters .
- a team of chefs work together to prepare a meal .
- a jar filled with liquid sits on a wood surface .
- a picture taken from the driver seat of car at a stop sign .
- a black and white cat sits near a window looking outside .

**MLE**

- here are motorcyclists on an trailer in front of folded .
- a man with a bandana is flipping
- a cat is resting on and pans and buttons of a bar and claw foot steel refrigerator has graffiti on
  the floor in it .
- a kitchen in a bench next to a tree in it
- a pretty orange parked on a table .

**LeakGAN**

- a group of people outside a jet airplane .
- a bathroom with a glass shower , sink , and tub .
- a black cat sitting nearby .
- a modern kitchen features microwave and refrigerator .
- a door showing the mirror in the reflection .

**MaliGAN**

- a man standing in a kitchen has an island with the stems
- a man standing in a giraffe cake .
- a car seat showing some break in a kitchen counter .
- a group of people that have ski down men standing next to a white accents .
- a toilet site in top in a neighborhood .

**RankGAN**

- a bathroom filled with sink , mirrors and white tile door .
- a bathroom with a large door with a blue window .
- a nice bathroom that is very clean table a yellow shirt and white umbrella at a stop .
- a bathroom with a shiny grey scheme a bathroom .
- a table with a flower dress is perched on it 's at a field .

**RelGAN**

- a bathroom with a white toilet next to a sink and a toilet paper roller in a bathroom .
- a red kitchen with glass doors open and red cupboards .
- a young boy cuts a cake designed to look like - a skateboard on the stove .
- a cat resting on top of a toilet seat in a bathroom .
- dirt bikers racing on a runway under the cabinet .

**SeqGAN**

- a template example of an airplane in a blue field
- small child looking at the crucifix on an umbrella on a plate .
- an older woman is sitting on a leash .
- the night of men is sitting up a very tall building .
- a close up of four planes on the street together .

**TaylorGAN**

- a group of people sitting around a police motorcycle .
- two people are near some airplanes are next .
- several white fixtures of a modern bathroom sink .
- an image of a woman in the kitchen .
- a herd of sheep grazing in the middle of a runway .

## Footnotes

[5]The preprocessed COCO dataset is available at https://github.com/pclucas14/GansFallingShort/tree/master/real_data_experiments/data/coco