[Reviews · NeurIPS 2020]

Review 1

Summary and Contributions: This paper proposed a novel update formula for GAN on natural language generation (NLG) task, which is called TaylorGAN. This model overcomes the difficulty of non-differentiable operation by the first-order Taylor expansion, which can incorporate the gradient information from the discriminator. Meanwhile, this paper also used discriminator constraint and entropy maximization to achieve reliable rewards and avoid mode dropping. The experimental results illustrated the higher performance of TaylorGAN than other comparative models.

Strengths: 1. Authors firstly used Taylor expansion to approximate the discrete outputs with the non-differentiable operation, and this method is able to alleviate the high variance and low sample efficiency problems caused by the sensitivity of sampling. 2. Apart from the Taylor estimator on discrete variables (tokens in NLG), authors also proposed the discriminator constraint to improve the rewards learning. The proposed reward is unbounded and doesn’t occur in the final non-linear transformation, so it’s coordinated with the Taylor estimator. 3. The experiment is reasonable and sufficient, and the results support well the claim of the effectiveness of the proposed approach.

Weaknesses: 1. As the Taylor estimator is a keystone in the proposed model, and the motivation is to improve the estimator for discrete random variable. This paper lacks the essential analysis and comparation between Taylor estimator and traditional methods, such as the Gumbel-Softmax and Gaussian-Softmax. 2. This paper lacks the data or experiment to verify the outperformance of Taylor estimator than other simplex on discrete random variable. I suggest authors to use the synthetic data to verify whether Taylor estimator has a more continuous modality than Gumbel-Softmax and others. 3. Authors didn’t make any case study to illustrate the generated text from TaylorGAN, even though they had compared the FED and LM scored in Table.1.

Correctness: yes

Clarity: yes

Relation to Prior Work: yes

Reproducibility: Yes

Additional Feedback:


Review 2

Summary and Contributions: To solve a high variance problem in reinforcement-based text generation, the work proposes a TaylorGAN, which uses the first-order Taylor expansion to approximate the reward of generated samples around their neighbor. The methodology is supported by solid explanation and theoretical basis. The experiment shows favorable results compared with previous GAN works and the MLE model.

Strengths: 1. The idea of using the first-order Taylor expansion to approximate the reward of partial generated sentence during is novel. 2. The methodology is supported by solid theoretical proof. The experimental results as well as the generated samples also look good. 3. Without the MLE pretraining, the proposed GAN model shows comparable or even better results than previous GAN models and MLE model.

Weaknesses: 1. Experiments should cover more datasets or tasks to prove the generalization of effectiveness, such as COCO captions with less average tokens or conditional generation tasks. Perplexity metric may have small problems.

Correctness: To my best knowledge, the method is correct (though I may miss some theoretical flaws).

Clarity: The writing is easy to follow.

Relation to Prior Work: The work should discuss more related works, especially the comparison with previous GAN works if space is allowed.

Reproducibility: No

Additional Feedback: Questions: 1. In equation (11), will the importance sampling of y also cause high-variance during the estimation? Is this also affected by the accuracy of neighborhood distribution (joint distribution) in equation 13? If yes, what's the performance if you use other types of kernel functions? 2. When you evaluate the perplexity, using your own model to calculate the perplexity may be problematic since it may favor the samples generated by itself. Why don't use pre-trained language models to evaluate the PPL? e.g., GPT-2 Other comments: 1. In line 103. It is helpful to elaborate more or mention the joint distribution will be introduced in 3.1.2, otherwise, the reader will confuse how do you construct this type of joint distribution. After the rebuttal: The author's responses solved my concerns. I have two strong recommendations for the author in the revision if the paper accepted by the conference: 1. Further clarify the questions raised in the reviews. Add more comparison discussions with previous text GANs. 2. Add more datasets (at least COCO captions dataset) in the experiment part. This will make the experiment more convincing.


Review 3

Summary and Contributions: This paper proposed a new GAN model for text generation utilizing Taylor Expansion.

Strengths: The idea of the algorithm is novel

Weaknesses: The evalation is not convincing. It is only tested on one dataset and the way to show the results are problematic. I highly recommend plot quality-diversity curve given temperature sweep. See this paper's evaluation as an example : https://arxiv.org/pdf/2004.13796.pdf

Correctness: See weakness.

Clarity: Yes.

Relation to Prior Work: Adequite. But text generation has various different applications, different evaluation metrics are weighted differently for different tasks. It seems the authors weren't too clear on this.

Reproducibility: Yes

Additional Feedback: I would want to see more datasets in evaluation and maybe do it only for unconditional generation but also for conditional generation.


Review 4

Summary and Contributions: The paper proposes a novel update formula for the generator in a GAN-based text generation model. The model uses Tylor expansion to quickly approximate the rewards of neighboring sentences, so that the generator can learn more efficiently. TaylorGAN achieves state-of-the-art performance without maximum likelihood pre-training. The proposed model can achieve low variance without additional variance reduction techniques.

Strengths: The idea of using the gradients in the discriminator to speed up the REINFORCE learning is a novel idea. And the idea is also intuitive as shown in Figure 1. The paper clearly specifies the distribution they pick and the derives the update formula. Experimental results show that TaylorGAN achieves state-of-the-art performance.

Weaknesses: This is a well written paper, but authors can further improve the paper by giving more intuitions and overviews before writing equations. For example, the first paragraph in section 3.1 can give a brief explanation of the algorithm like “”” In our model, when we sample a sentence x and get its reward R, we can efficiently approximate neighboring sentences using first order Tylor expansion of R. Estimating more sentences can reduce variance. But we have to solve xxxx challenges. “”” Also consider refer to Figure 2 and explain the figure in this paragraph. Line 123 "we ignore the effect of v on the rest of auto-regressive sampling. " I understand it is a necessary trade-off. But in natural language, substitute one word can change the distribution of the following words a lot. Authors can consider discuss the potential effect of this approximation. The authors do not provide their implementation, which can increase the difficulty of reproduction.

Correctness: likely to be correct.

Clarity: This is a well-written paper. The related works, the method and experiments is clearly explained.

Relation to Prior Work: Yes.

Reproducibility: Yes

Additional Feedback: In Table 2, consider filling in all the omitted numbers. Line 71 samples a new token x_t ***from*** a soft policy \pi Line 133 remind reader that \Lambda appears in Eq (13). 
 In the examples shown in the appendix, the sentences generated by TaylorGAN are shorter than other methods. Is it an expected behavior? Would sentence length affect the evaluation metrics? If space allows, please add a few generated examples to the main paper.

[Author Response · NeurIPS 2020]

Table 1: FED, LM score & perplexity on EMNLP 2017 News. Lower is better. Addition to metrics in the submitted paper, we evaluate perplexity on the self-generated samples to verify the correctness. The discriminator constraints and the entropy maximization are used in all experiments.

| Estimator | FED | | Perplexity | | |
|---|---|---|---|---|---|
| | Train | Val. | Self | Train | Val. |
| Gumbel-Softmax | 0.0141 | 0.0218 | 13 | 5267 | 6369 |
| REINFORCE | 0.0117 | 0.0182 | 25 | 68 | 72 |
| Taylor | 0.0105 | 0.0149 | 26 | 67 | 72 |

Figure 1: Sentence length statistics.

## 1 Reviewer #1

• **Response to Weakness 1, 2**: Regarding traditional estimators for discrete random variables, there are comparisons
with REINFORCE and straight-through in Table 2[†], discussions in line 133[†] and Appendix C[†]. Gumbel-Softmax
was reported with bad performance in Zhu *et al.* [3], which is confirmed in our experiments. As shown in Table 1,
Taylor estimator outperforms Gumbel-Softmax on FED and perplexity. We argue that the inferior performance is the
consequence of biased and spiky distribution explained in both line 47-51[†] and the unusually high perplexity on real
data, even with temperature annealing during the training phase [2].

## 8 Reviewer #2

• **Response to Question 1**: If the estimation is performed with Monte-Carlo sampling on $\mathbf{y}$ by a bad proposal $\Gamma$,
the variance can be high. However, in equation (11)[†], the gradient is calculated by *summing* up all $\mathbf{y}$s instead of
sampling. Therefore, the variance is reduced by incorporating more samples if $\Gamma$ is uniform enough. The choice of
distribution is *arbitrary* (if computed efficiently), so there is no such "target" distribution to be fit or accuracy concern
in equation (13)[†]. We have experimented kernels such as Epanechnikov and tricube, but none of them outperforms
Gaussian. Which kernel function achieves the best balance between bias and variance is an interesting question, and
we leave the theoretical investigation to future study.

• **Response to Question 2**: We evaluate the results with perplexity as well as language model scores. The perplexity
in Table 2[†] refers to the inverse of the per-word probability of the model generating the *validation data*, which
is independent of the generated samples. We guess the metric you mentioned may be the "LM score" defined in
line 199[†], for which we indeed trained another model following the settings of de Masson d'Autume *et al.* [1],
mentioned in line 198[†].

## 21 Reviewer #3

• **Response to Weakness**: The quality-diversity curve given temperature sweep is plotted in Figure 3[†], which is
mentioned in line 219[†]. We will emphasize this in the caption of Figure 3[†] in the final submission.

## 24 Reviewer #4

• **Response to Question**: The shorter samples in Table 4[†] are not a expected behavior but a coincidence. TaylorGAN
does not tend to generate shorter sentences, as shown in Figure 1. de Masson d'Autume *et al.* [1] has found some
correlation between the sentence length and the FED score and then designed their model accordingly. We, on the
other hand, do not utilize this correlation. Among our metrics, the BLEU score penalizes short sentences by an
exponentially decaying term.

## 30 References

[1] C. de Masson dAutume, S. Mohamed, M. Rosca, and J. Rae. Training language gans from scratch. In H. Wallach,
H. Larochelle, A. Beygelzimer, F. dAlché-Buc, E. Fox, and R. Garnett, editors, *Advances in Neural Information*
*Processing Systems 32*, pages 4300–4311. Curran Associates, Inc., 2019.

[2] E. Jang, S. Gu, and B. Poole. Categorical reparameterization with gumbel-softmax, 2016.

[3] Y. Zhu, S. Lu, L. Zheng, J. Guo, W. Zhang, J. Wang, and Y. Yu. Texygen: A benchmarking platform for text
generation models. In *SIGIR on Research & Development in Information Retrieval*, pages 1097–1100, 2018.

† refers to the submitted paper and the supplementary material.

[Meta-Review · NeurIPS 2020]

This paper proposed a novel method for GAN-based natural language generation, where first order Taylor expension is used to estimate the gradient of the reword function. This method greatly mitigate the high variance problem of previous methods and improve the sample efficiency. Experiments show the proposed method achieve the state-of-the-art. The work is solid both in theory and in experiments.